# Treatment of Severe Atopic Dermatitis with Dupilumab in Three Patients with Renal Diseases

**DOI:** 10.3390/life12122002

**Published:** 2022-11-30

**Authors:** Caterina Foti, Paolo Romita, Francesca Ambrogio, Carlo Manno, Raffaele Filotico, Nicoletta Cassano, Gino Antonio Vena, Aurora De Marco, Gerardo Cazzato, Biagina Gisella Mennuni

**Affiliations:** 1Section of Dermatology, Department of Biomedical Sciences and Human Oncology (DIMO), University of Bari “Aldo Moro”, 70124 Bari, Italy; 2Section of Nefrology, Department of Emergency and Organ Transplantation (DETO), University of Bari “Aldo Moro”, 70124 Bari, Italy; 3Dermatology and Venereology Private Practice, 76121 Barletta, Italy; 4Dermatology and Venerology Private Practice, 70100 Bari, Italy; 5Section of Pathology, Department of Emergency and Organ Transplantation (DETO), University of Bari “Aldo Moro”, 70124 Bari, Italy

**Keywords:** atopic dermatitis, dupilumab, Alport syndrome, IgA nephropathy

## Abstract

Background: Atopic dermatitis (AD) is a chronic relapsing inflammatory skin disease that can affect patients’ quality of life. Dupilumab is the first biologic agent approved for the treatment of patients with inadequately controlled moderate-to-severe AD and its mechanism of action is based on the inhibition of the interleukin (IL)-4 and IL-13 signaling. There are only a few data on the safety of dupilumab in AD patients with comorbidities, including kidney disorders. Materials and Methods: Descriptive retrospective series of three patients with chronic kidney diseases (Alport syndrome, IgA nephropathy, and hypertensive nephrosclerosis, respectively) receiving dupilumab for their concomitant severe AD. Results: Treatment with a standard dosage of dupilumab caused a relevant improvement of AD in all patients without any adverse events or worsening of renal function. In a patient with severe renal failure, the drug was effective and well tolerated without the need for any dose adjustments, also after the initiation of peritoneal dialytic treatment. Conclusion: Our case series suggests the use of dupilumab as an effective and safe treatment for AD patients suffering from renal diseases, although additional studies are required to confirm such preliminary findings.

## 1. Introduction

Atopic dermatitis (AD) is a chronic relapsing inflammatory skin disease that can have a significant impact on the quality of life of affected patients and their families. Dupilumab is a human monoclonal antibody that inhibits interleukin (IL)-4 and IL-13 signaling by binding the shared alpha-subunit of the IL-4 receptor, thus reducing the Th2 response [1]. Dupilumab is the first biologic agent that has been approved for the treatment of patients with inadequately controlled moderate-to-severe AD. There are only scant data on the safety of dupilumab and other systemic therapies in AD patients with comorbid conditions, including kidney disorders, as well as elderly patients, also because such patients are usually excluded from randomized clinical trials [2].

We describe three male adult patients with severe AD and kidney disease who were safely and successfully treated with a standard dosage of dupilumab (600 mg induction dose followed by 300 mg every 2 weeks thereafter).

## 2. Report of Cases

The first patient reported a long personal history of allergic rhinitis, asthma and severe AD, secondary arterial hypertension, and chronic kidney disease due to Alport syndrome, a rare hereditary disease caused by mutations in the type IV collagen genes COL4A3, COL4A4, and COL4A5, affecting the renal glomerular basement membrane [3]. A reduction in the Eczema and Severity Index (EASI) of at least 90% from baseline (EASI 90) was reached after 16 weeks of treatment with dupilumab. One year after treatment initiation, an almost complete clearance of AD lesions and a slight improvement of renal function were noticed (Table 1). These effects persisted after two years of continuous treatment.

The second patient, affected by IgA nephropathy, reported a 10-year history of pollen allergy and a 4-year history of severe AD. AD manifestations rapidly improved with dupilumab leading to the achievement of EASI 90 in two months. After one year, a consistent reduction in AD scores was registered, while serum creatinine and glomerular filtration rate did not show remarkable changes; a slight reduction in proteinuria was observed (Table 1). Such results were maintained after 34 months of continuous therapy with dupilumab.

The third patient was an elderly subject with severe AD, chronic kidney disease probably due to hypertensive nephrosclerosis, and chronic obstructive pulmonary disease. After one year of therapy with dupilumab, AD scores dramatically improved and renal function remained stable (Table 1). Clinical results after 16 weeks of therapy are shown in Figure 1. However, due to the severity of the renal disease, peritoneal dialysis was initiated one year after dupilumab initiation, without the need for any dosage adjustments. During a 12-month follow-up period following the start of peritoneal dialysis, dupilumab treatment continued to be effective in controlling AD and did not induce any untoward effects or further deterioration of renal function.

All patients applied emollients and used short courses of topical corticosteroids on an as-needed basis, especially in the initial treatment phases.

Table 1 summarizes the main characteristics of patients and results after one year of treatment with dupilumab.

## 3. Discussion

Dupilumab proved to be safe and effective in these three patients with severe AD and concomitant kidney disease, without any adverse events or worsening of renal function during treatment. Moreover, in the third patient affected by severe renal failure, the renal function remained stable during treatment with dupilumab, and no dose adjustments were required, even after the initiation of peritoneal dialytic treatment.

Only a few publications described the use of dupilumab for AD in patients with severe systemic comorbidities, including renal diseases [2,4]. However, the effects of dupilumab treatment on renal function or detailed safety data have been specified in a limited number of reports, mainly regarding isolated cases.

Renal disease is an obvious contraindication to cyclosporine, one of the most common therapeutic options for severe AD. Moreover, patients with chronic kidney disease, especially those with end-stage renal failure or on renal replacement treatment, frequently complain of chronic pruritus [5]. Therefore, concurrent renal disorder may influence itch intensity in patients suffering from concurrent AD or other pruritic dermatoses. Interestingly, Zhai et al. reported the response of various pruritic conditions to dupilumab, documenting the improvement of symptoms in five patients with uremic pruritus, one of whom in hemodialytic therapy [6].

Kha et al. described a renal transplant man with uremic pruritus, acute flare of AD and superimposed prurigo nodularis coinciding with recent transplant failure who was treated with dupilumab for 8 months [7]. Treatment was efficacious and well tolerated, and changes in the concurrent immunosuppressive therapy were not necessary. Similarly, another report demonstrated that dupilumab was safe and effective for severe and recalcitrant AD in an immunosuppressed renal transplant patient with an unspecified genetic renal syndrome [8].

Varma et al. presented the case of a 22-month-old child with a history of hydronephrosis and other pathological conditions, whose AD dramatically improved after receiving three doses of dupilumab [9]. A recent publication outlined the positive outcomes obtained with dupilumab in two children with AD and nephrotic syndrome [10].

Anecdotal and contrasting findings exist regarding the safety of dupilumab in patients with IgA nephropathy. IgA nephropathy is the most common primary glomerulonephritis in the world and is characterized by the deposition of IgA1-containing immune complexes in the glomerular mesangium responsible for glomerular injury. Pathogenesis of IgA nephropathy is only partially known and appears to be complex. Excessive activity of T lymphocytes, especially the Th2, Tfh, Th17, and Th22 subpopulations, is likely to have a relevant pathogenic role [11]. Moreover, it has been speculated that IgE might be involved in the pathogenesis of IgA nephropathy and serum IgE level might be associated with renal progression [12].

In our patient with IgA nephropathy, a slight reduction in proteinuria was detected during treatment with dupilumab. IgE levels were only slightly above the normal range at baseline and marginally decreased after one year of dupilumab treatment (Table 1). Tanczosova et al. reported similar positive results in a patient with IgA nephropathy whose severe AD worsened during prednisone tapering [13]. In their patient, after treatment with dupilumab for 9 months in association with a minimum dose of prednisone, such authors observed the achievement of EASI 90, an improvement of renal laboratory parameters, and a substantial decrease of serum IgE levels (from the baseline value of 12,500 IU/mL up to 3490 IU/mL). Tanczosova et al. hypothesized that reasons for the improved renal function after dupilumab therapy could be the inhibition of systemic inflammation and the reduction of IgE concentrations [13].

In another report (14), a relevant decline in serum IgE levels and an unexpected rapid deterioration in renal function were described during treatment with dupilumab in a patient with severe AD and minimal renal dysfunction. After stopping dupilumab, a renal biopsy was performed, and primary IgA nephropathy was diagnosed. The authors of this report suspected the involvement of IL-4 neutralization in IgA nephropathy exacerbation through the possible activation of Th17 cells or Toll-like receptor signaling [14]. Therefore, the role of dupilumab in IgA nephropathy is still controversial and requires further investigations.

In conclusion, the present cases suggest that dupilumab can be an effective and safe option in AD patients suffering from renal diseases. However, more data are necessary to better define the safety profile in subjects with relevant comorbidities and special patient populations, including patients with chronic kidney disease and subjects treated with dialysis.

## Figures and Tables

**Figure 1 life-12-02002-f001:**
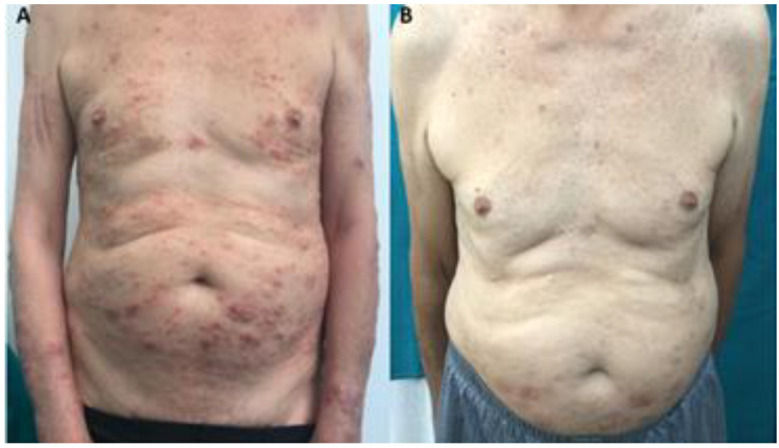
Patient 3 at T0 (**A**) and after 16 weeks of therapy with dupilumab (**B**).

**Table 1 life-12-02002-t001:** Characteristics of three male patients with severe AD and chronic kidney disease and their laboratory parameters at baseline (T0) and after one year of treatment with dupilumab (T1).

	Patient 1		Patient 2		Patient 3	
Age at Baseline (yrs)	36		44		77	
**Renal disease**	Alport syndrome		IgA nephropathy		Hypertensive nephrosclerosis	
**Concomitant treatment**	Ramipril		Ramipril		Nifedipine	
**Previous treatmentsfor AD ^**	Emollients, topical corticosteroids, phototherapy		Emollients, topical corticosteroids		Emollients, topical corticosteroids	
	**T0**	**T1**	**T0**	**T1**	**T0**	**T1**
**EASI score**	31	1	27	0	37.5	6
**P-NRS**	7	2	7	0	10	4
**IGA score**	4	1	4	0	4	1
**DLQI score**	20	1	15	0	25	2
**Serum IgE (IU/mL)**	50.1	40	145	58.7	103	55
**Eosinophils (%)**	5.1	4.5	2.1	2	13.3	10.7
**Serum creatinine (mg/dL)**	2.15	1.87	0.90	1.05	4.85	4.94
**eGFR (mL/min/1.73 m^2^)**	38	43	97	86	12	12
**Proteinuria (g/24 h)**	2.40	2.50	1.61	1.39	1.2	1.1

^ Phototherapy was refused by patient 2 because of difficulties with work schedule and by patient 3 because of problems related to travel time and distance. AD, atopic dermatitis; DLQI, Dermatology Life Quality Index; EASI, Eczema Area and Severity Index; eGFR, estimated glomerular filtration rate; IGA, Investigator’s Global Assessment; P-NRS, pruritus numerical rating scale. Normal range of laboratory parameters: eosinophils 0.3–6.2%; serum IgE < 100 IU/mL; eGFR > 90 mL/min/1.73 m^2^; serum creatinine < 1.20 mg/dL; proteinuria < 0.20 g/24 h.

## Data Availability

Not applicable.

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
