# Peer review of "Treatment of Severe Atopic Dermatitis with Dupilumab in Three Patients with Renal Diseases"

_life, 2022, doi:10.3390/life12122002_

Round 1
Reviewer 1 Report
Dupilumab in kidney disease has been shown to be safe. The decrease in proteinuria is also linked to IL13, secondary to the regulatory role of IL4.
The authors should include a table with previous treatments, I am surprised that they have not received previous phototherapy, perhaps because they do not have it.
A table with the evolution of the EASI, IGA, and pruritus scale. Include previous treatments in the case description section.
Author Response
Reviewer 1
The authors should include a table with previous treatments, I am surprised that they have not received previous phototherapy, perhaps because they do not have it.
A table with the evolution of the EASI, IGA, and pruritus scale. Include previous treatments in the case description section.
Re:Thanks for your comment. As suggested, we have included the evolution of EASI, IGA and P-NRS, and previous treatments for atopic dermatitis in the table. As regards phototherapy, only one patient was treated with phototherapy because patients 2 and 3 refused it as specified in the table footnotes.
Reviewer 2 Report
The present manuscript describes treatment result of severe atopic dermatitis with Dupilumab, an antibody drug against IL-4 alpha subunit, in three patients with Alport syndrome, IgA nephropathy and hypertensive nephrosclerosis. After one year treatment of Dupilumab EASI score, P-NRS and DLQI score were remarkably improved without adverse effects. The present case report shows effective and safe treatment of Dupilumab for three atopic dermatitis patients with severe renal disease. Although additional survey remains to be performed, the present study is thought evaluated. Then the present manuscript is thought acceptable to “Life” journal.
Author Response
Reviewer 2
The present manuscript describes treatment result of severe atopic dermatitis with Dupilumab, an antibody drug against IL-4 alpha subunit, in three patients with Alport syndrome, IgA nephropathy and hypertensive nephrosclerosis. After one year treatment of Dupilumab EASI score, P-NRS and DLQI score were remarkably improbe without adverse effects. The present case report shows effective and safe treatment of Dupilumab for three atopic dermatitis patients with severe renal disease. Although additional survey remains to be performed, the present study is thought evaluated. Then the present manuscript is thought acceptable to “Life” journal.
Re:We are extremely flattered by this comment and we thank you very much. We would like to follow our patients over time. We agree with you that further research and additional data on large patient samples are required
Reviewer 3 Report
This case series focused on the safety of Dupilumab in kidney diseases.
1 Severe atopic dermatitis in the three patients could be shown by pictures.
2 The authors mainly discussed the possible mechanism of Dupilumab in IgAN. There have been two cases reported Dupilumab in IgAN, but one of them found exacerbation of IgA nephropathy. Therefore the role of Dupilumab in IgAN may be controversial.
3 Table 1 could be reorganized with T0 and T1 in separate rows.
Author Response
Reviewer 3
1 Severe atopic dermatitis in the three patients could be shown by pictures.
2 The authors mainly discussed the possible mechanism of Dupilumab in IgAN. There have been two cases reported Dupilumab in IgAN, but one of them found exacerbation of IgA nephropathy. Therefore the role of Dupilumab in IgAN may be controversial.
3 Table 1 could be reorganized with T0 and T1 in separate rows.
Re: Thank you for your suggestions. We have included a figure in which we show patient 3 before and after 16 weeks of treatment with dupilumab. We considered in the discussion the controversial role of dupilumab in IgA nephropathy and we concluded that it requires further investigations. We also reorganized the table, hoping that this format will make it easier to read.
Kind regards
Round 2
Reviewer 3 Report
My comments are solved.